

# GNSS Radio Occultation Climatologies mapped by Machine Learning and Bayesian Interpolation

Endrit Shehaj[1,2], Stephen Leroy[3], Kerri Cahoy[1], Alain Geiger[2], Laura Crocetti[2], Gregor Moeller[2], Benedikt Soja[2], Markus Rothacher[2]

[1]STAR lab, Department of Aeronautics and Astronautics, Massachusetts Institute of Technology, Cambridge, MA 02139, USA
[2]Institute of Geodesy and Photogrammetry, ETH Zürich, Zürich, 8093, Switzerland
[3]Atmospheric and Environmental Research, Lexington, MA 02421, USA

*Correspondence to*: Endrit Shehaj (endrit.shehaj@geod.baug.ethz.ch)

**Abstract.** Global Navigation Satellite Systems (GNSS) radio occultation (RO) is a space-based remote sensing technique that measures the bending angle of GNSS signals as they traverse the Earth's atmosphere. Profiles of the microwave index of refraction can be calculated from the bending angles. High accuracy, long-term stability, and all-weather capability make this technique attractive to meteorologists and climatologists. Meteorologists routinely assimilate RO observations into numerical weather models. RO-based climatologies, however, are complicated to construct as their sampling density is highly non-uniform and too sparse to resolve synoptic variability in the atmosphere.

In this work, we investigate the potential of machine learning (ML) to construct RO climatologies and compare the results of a ML construction with Bayesian interpolation (BI), a state-of-the-art method to generate maps of RO products. We develop a feed-forward neural network applied to COSMIC-2 RO observations and simulate data taken from the atmospheric analyses produced by the European Centre for Medium-Range Weather Forecasts (ECMWF). Atmospheric temperature, pressure and water vapor are used to calculate microwave refractivity at 2, 3, 5, 8, 15, and 20 km geopotential height, with each level representing a different dynamical regime of the atmosphere. The simulated data are the values of microwave refractivity produced by ECMWF at the geolocations of the COSMIC-2 RO constellation, which fall equatorward of 46° latitude. The maps of refractivity produced using the neural networks better match the true maps produced by ECMWF than maps using BI. The best results are obtained when fusing BI and ML, specifically when applying ML to the post-fit residuals of BI. At the six iso-heights, we obtain post-fit residuals of 10.9, 9.1, 5.3, 1.6, 0.6 and 0.3 *N*-units for BI and 8.7, 6.6, 3.6, 1.1, 0.3 and 0.2 *N*-units for the fused BI&ML, respectively. These results are independent of season.

The BI&ML method improves the effective horizontal resolution of the posterior longitude-latitude refractivity maps. By projecting the original and the inferred maps at 2 km iso-height onto spherical harmonics, we find that the BI-only technique can resolve refractivity in the horizontal up to spherical harmonic degree 8 while BI&ML can resolve maps of refractivity using the same input data up to spherical harmonic degree 14.



## 1 Introduction

Earth radio occultation (RO) sounds temperature and water vapor in the Earth's atmosphere by measuring the refraction-induced frequency-shifting of the signals of the Global Navigation Satellite Systems (GNSS) satellites as received by satellites in low-Earth orbit (LEO). The RO remote sensing technique has been thoroughly described in several previous

works, e.g., (Kursinski, et al., 1997), (Kursinski, et al., 2000), (Melbourne, 2004), (Mannucci, et al., 2021). As an active limb-sounding technique, it provides highly accurate information on temperature and water vapor with 100-m vertical resolution from the surface to the stratopause, but with non-homogenous, non-uniform and sparse horizontal sampling. In order to convert the soundings to a gridded dataset, special algorithms must be devised to map the data in the horizontal. The horizontal sampling is non-homogeneous because the orbital configuration of the multiple RO spacecraft have not been

coordinated, and due to variation in occultation measurement opportunities when combined with the orbital motion of the GNSS satellite constellations. Consequently, local time coverage is generally incomplete and meridional coverage at times neglected. Because of orbital dynamics, gaps in the RO sampling pattern occur at specific, well-defined latitudes (Leroy, et al., 2012). The horizontal sampling distribution is also sparse because the density of RO soundings has rarely been high enough to sample every cell of synoptic variability in the atmospheric system, where a cell is approximately described by a

span of several hours and the spatial Rossby radius of deformation. Mapping RO data requires statistical methods that weight the RO observations that do exist in a manner that minimizes the errors incurred by under-sampling the atmosphere without inducing biases.

Two approaches have been developed to construct gridded RO climatologies. The first approach is referred to as "sampling-error-removal". It uses the forecasts of a numerical weather prediction (NWP) model to estimate the sampling

error associated with synoptic variability and incomplete coverage of the diurnal cycle. In this approach, the forecasts of an NWP system are interpolated to the locations and times of RO soundings, binned and averaged into longitude-latitude boxes just as the actual RO soundings are, and compared to the gridded predictions to estimate a bias associated with the binning and averaging. This estimate of the bias is then subtracted from the actual binned-and-averaged RO data (Foelsche, et al., 2008), (Foelsche, et al., 2011). The other approach is Bayesian interpolation (BI) on a sphere, wherein linear combinations of

spherical harmonics as basis functions are fit to RO data without overfitting the data (MacKay, 1992), (Leroy, et al., 2012). There have been many other applications of BI, and it has been evaluated in detailed analyses as a method for constructing climatologies of RO data (Leroy, et al., 2012), (Leroy, et al., 2021).

In this work, we use machine learning (ML) to produce RO climatologies. The ability of ML to learn from large amounts of data has been shown in many research subjects and applications (Hassanien, 2018). Neural networks are well suited to the

problem of estimating most probable values related to generalized inputs, which is the same problem that affects gridding RO data in the horizontal. Unlike the sampling-error-removal approach, neither BI nor ML require external datasets to form objective gridded climatologies of RO data. At its core BI superposes spherical harmonics, while ML is based on more general mathematical functions. Both approaches are appropriate for exploiting large amounts of data.



ML has already been deployed successfully on ground-based GNSS observations. For decades, parameters estimated from GNSS data (such as station coordinates, troposphere, ionosphere, etc.) have been routinely quantified at permanent geodetic stations. In addition, the size and resolution of GNSS networks have increased, following the requirements of meteorologists, geodesists and geophysicists. Thus, a large amount of data has been produced, which has naturally generated interest in applying ML algorithms to these datasets. (Kiani Shahvandi, et al., 2022), (Gou, et al., 2023) and (Natras, et al., 2022) applied ML to improve the prediction of important parameters needed in GNSS applications, such as polar motion prediction, ultra-rapid orbits, and ionosphere, while (Crocetti, et al., 2021) used ML to detect discontinuities in time series of GNSS station coordinates.

ML algorithms have been successfully used to model meteorological products derived from GNSS observations in the past. For instance, in (Miotti, et al., 2020) and (Shehaj, 2023), ML was applied to tropospheric observations of ground-based GNSS to model them based on meteorological parameters. (Miotti, et al., 2020) showed that ML could model the implicit relation between zenith total delays (ZTDs), estimated at ground-based GNSS stations and meteorological parameters measured at permanent meteorological stations. While (Miotti, et al., 2020) demonstrated the applicability of ML to map time series of GNSS tropospheric observations, in (Shehaj, et al., 2022) ML and least-squares collocation were combined to produce high-resolution ZTD fields.

In other work, ML has been applied to tropospheric delays estimated at GNSS ground-based stations for prediction of Alpine Foehn, (Aichinger-Rosenberger, et al., 2022), or to spatially map zenith wet delay at a global scale, (Crocetti, et al., n.d.). (Kitpracha, et al., 2019) used LSTM and a combination of singular spectrum analysis with Copula to predict zenith delays based on previous meteorological and delay series; errors of 2 cm and 1 cm were reported for a prediction of 24 hours. In (Shamshiri, et al., 2019), a ML Gaussian Process to model tropospheric delays in InSAR based on zenith delays was used, reporting an improvement of 81% on the tropospheric corrections of the interferograms. In (Zhang & Yao, 2021), ML was applied to fuse precipitable water vapor from GNSS, MODIS (Moderate-Resolution Imaging Spectroradiometer) and the numerical weather model ERA5. In (Shi, et al., 2023) a method to efficiently generate zenith delays for the massive GNSS CORS (continuously operating reference station) network utilizing ML is developed.

While in previous work ML was successfully deployed to map time series of GNSS ground-based atmospheric observations, in this work we apply ML for spatial and temporal mapping of GNSS RO measurements. We exploit the large number of RO measurements (thousands daily) and the ability of ML to learn patterns from large datasets.

We develop a neural network for interpolating RO data in 6-hr cycles in order to create gridded climatologies and compare the results to maps generated using BI. We can only compare our ML approach to BI since it provides a-posterior uncertainty, but the sampling-error-removal method does not. We also estimate performance with simulation-mapping experiments using the output of an NWP system as a nature run. In doing so we treat the gridded model output as 'truth' against which we compare the output of the various mapping methods we consider. The outcomes are estimates of the uncertainty and the performance of each mapping approach.



The second section of this paper describes the data that we use. The third section describes the BI and ML mapping algorithms. The fourth section contains the analysis of a numerical experiment that probes the performance of BI and ML. Finally, the fifth section presents a summary discussion of results and future work.

**2 Data**

**2.1 COSMIC-2/FORMOSAT-7**

The COSMIC-2/FORMOSAT-7 mission is operated by National Oceanic and Atmospheric Administration (NOAA), US AIR Force (USAF), Taiwan's National Space Organization (NSPO), University Consortium for Atmospheric Research (UCAR) and other partners (UCAR, 2022), (Ho, et al., 2020), (Schreiner, et al., 2020). At present, COSMIC-2 obtains RO 105 soundings from the transmitters of the U.S. Global Positioning System (GPS) and the Russian GLONASS, providing approximately 6,000 high-performance profiles of refractivity daily and covering the Earth from 46°S to 46°N latitude. We use microwave refractivity wetPf2 from the data portal of the COSMIC project office of UCAR. The wetPf2 NetCDF files contain geometric altitude above mean sea level, geopotential height above mean sea level, longitude, latitude, temperature, pressure, water vapor partial pressure, specific humidity, relative humidity, dry temperature, dry pressure and refractivity for 110 each level of the atmospheric sounding. In this work, we use geometric altitude above mean sea level, longitude, latitude and refractivity.

We interpolate the COSMIC-2 refractivity profiles to isohypsic surfaces 2, 3, 5, 8, 15, and 20 km above mean sea level. These levels show a wide variety of morphologies in spatial-temporal structures because of the very different physical phenomena prevalent at each level:

- 2 km: at this height, we notice small-scale structures (see Figure 1) related to boundary layer clouds and water vapor.
- 3 km: at this height, there is still an important contribution of the water vapor to refractivity, but it is just outside the planetary boundary layer. We expect the retrieved refractivity to have higher quality than at 2 km, since Abel inversion for refractivity encounters its largest errors within the boundary layer, usually associated with super-refraction and tracking difficulties.
- 5 km and 8 km: synoptic, jet stream, and frontal variability dominate the dynamics of refractivity with a smaller contribution of water vapor than in the boundary layer.
- 15 km: mixing across the subtropical front by baroclinic eddies in the stratospheric middle world dominates. In the mid-latitudes, we are in the stratosphere, while in the tropics we are in the troposphere. This is depicted in Figure 1, where a clear distinction – almost a step function – in refractivity is experienced between the tropics and mid-latitudes.
- 20 km: larger structures of the atmosphere related to planetary scale waves in the lower stratosphere.





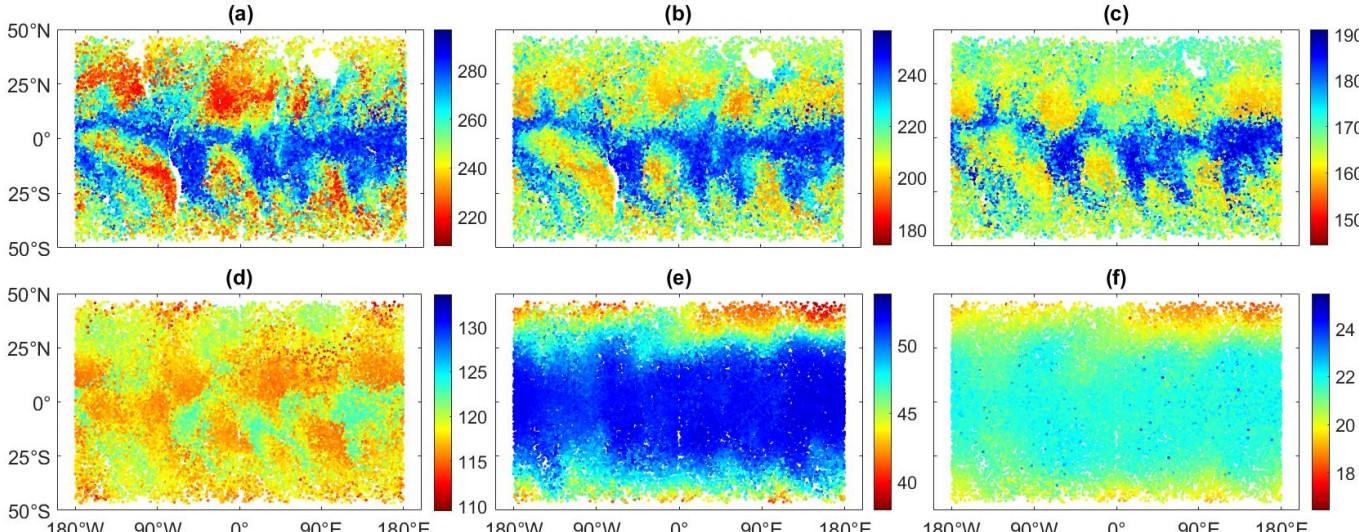

**Figure 1: COSMIC-2 RO refractivity distributions on six isohypsic surfaces for the period 1–10 January 2020, illustrating the very different spatial-temporal morphologies at each level. The refractivity at 2, 3, 5, 8, 15 and 20 km is visualized in the plots (a), (b), (c), (d), (e) and (f), respectively. The colorbar units are N-units (ppm), i.e., refractivity units, the x-axis is east longitude, and the y-axis is northward latitude.**

We utilize COSMIC-2 data representing 10-day time series of the four seasons. The measurements spanning 1–10 January 2020 represent Boreal winter (40,000 profiles), 1–10 April 2020 represent Boreal spring (40,000 profiles), 3–12 July 2020 represent summer (30,000 profiles), and 1–10 October 2020 represent fall (30,000 profiles). The criterion to select these timespans was continuity, meaning that we simply chose the first 10 continuous days with RO profiles for each season of 2020.

### 2.2 ECMWF Operational Forecasts

In order to create a "nature run" on which to test ML mapping schemes, we interpolated the forecasts of the operational weather prediction system of the European Centre for Medium-range Weather Forecasts (ECMWF) to the times and locations of COSMIC-2 RO soundings, spanning 1–10 January 2020. Specifically, we used the output of ECMWF Integrated Forecast System (IFS) cycles 46r1 and 47r1; see (ECMWF, 2023). By using forecast fields rather than analysis fields, complications that arise from assimilating COSMIC-2 RO data into ECMWF operational analyses are avoided. Forecasts are always physically consistent three-dimensional fields of the atmosphere in as much as the physics is defined by the prognostic model. NWP analyses, however, are physically inconsistent because the data that constrain the atmospheric state perturb the state in isolated regions away from physical consistency. We obtained NetCDF files of pressure, temperature, water vapor, and geopotential fields with a horizontal resolution of 0.5°. We used 12-hr forecast fields, which are published hourly.



We computed refractivity profiles at the times and locations of COSMIC-2 RO soundings at the grid points in operational forecasts and interpolated linearly in time. Refractivity $N$ is related to the microwave index of refraction $n$ and atmospheric properties, (Rueger, 2002):

$$N = (n-1) \times 10^6 = (77.6890 \text{ K hPa}^{-1})\frac{(p-p_w)}{T} + (71.2952 \text{ K hPa}^{-1})\frac{p_w}{T} + (375463 \text{ K}^2 \text{ hPa}^{-1})\frac{p_w}{T^2}, \qquad (1)$$

where $p, p_w, T$ are the atmospheric pressure, the partial pressure of water vapor, and temperature, respectively. This formulation of refractivity accounts for fixed and induced dipoles of nitrogen, oxygen, carbon dioxide and water vapor, but neglects compressibility effects. The refractivity of surface air generally falls in the interval 320 to 360 $N$-units, about 10% of which is due to water vapor, with larger values in lower latitudes where more water vapor is present. When interpolating the model to the times and locations of COSMIC-2 RO soundings, we took the model refractivity profile in the cell nearest to the RO sounding and interpolated linearly in time and linearly in altitude, the vertical dimension.

Finally, we interpolated the refractivity for each profile to the six chosen altitudes listed in section 2.1. Using ECMWF data with resolutions of 0.5° in latitude/longitude and 1-hour in time, we locate the closest forecast geolocations and times of the COSMIC-2 RO data.

## 3 Methods

We introduce two RO mapping techniques in this section: Bayesian interpolation on a sphere, and machine learning via neural networks.

### 3.1 Bayesian interpolation (BI)

Bayesian interpolation (BI) works by fitting irregularly gridded and noisy data by an expansion of basis functions without overfitting the data, (MacKay, 1992). The input to the method is a set of scalar values with associated longitudes and latitudes and possibly even local (solar) times. The output is an inference of a two-dimensional field and its uncertainty as the coefficients of a spherical harmonic expansion and sines and cosines in the diurnal cycle along with an associated uncertainty covariance matrix of those coefficients. Diagnostics of the process yield information on the effective degrees of freedom of signal—and hence the horizontal resolution of the map—a single value describing the "measurement" error of every input value, and the Bayesian evidence of the fit, otherwise known as the joint probability of the model and the data. While largely an objective method, it nevertheless does involve some tuning of the regularization matrix, the purpose of which is to prevent overfitting of the data. We use a regularization matrix that asymptotes to stable values of Bayesian evidence with increasingly large spherical harmonic expansions (Leroy, et al., 2012). In doing so, we assure that output mappings are neither penalized nor rewarded for increasing numbers of basis functions beyond some nominal expansion.

BI is well suited to map GNSS RO data because the sampling patterns are highly irregular and synoptic variability acts as a source of noise (Leroy, 1997). On a sphere, the natural basis functions are spherical harmonics, and BI using spherical



harmonics as basis functions has been explored in depth to generate level 3 climatologies (i.e. latitude-longitude gridded products) of GNSS RO data (Leroy, et al., 2012), (Leroy, et al., 2021). For this work, we use the same Python module developed by (Leroy, et al., 2012), (Leroy, et al., 2021). We map RO observations in latitude and longitude and use the BI results to compare and combine them with ML. While BI on a sphere is intended for globally distributed, nonuniformly sampled data, it also works well when the data are restricted geographically. In our application, COSMIC-2 RO sounding distributions are restricted to the tropics and to the oceans. An example of a BI map is shown in Figure 5, plot (b).

**3.2 Machine learning applied to RO**

We use the classical artificial neural networks algorithm Multilayer Perceptrons (MLPs) detailed in (Haykin, 2009). This algorithm is widely applied for large datasets. We apply a fully connected neural network, where the neurons of one layer are connected to all the neurons of the previous one. The first layer consists of the inputs and the last one of the target values. Each neuron is computed as follows (Haykin, 2009):

$$neuron = f(\sum_{i=1}^{n} neuron\_previous\_layer_i * w_i + b) \tag{2}$$

where a weight $w_i$ is computed for each neuron of the previous layer, and a bias $b$ is added. The bias and weights are the parameters of the neural network.

The inputs and outputs have complex relations. This might require some complex nonlinear equations. The activation function $f$ defines the nonlinearity. The most common activation function is the Rectified Linear Unit function (ReLU); it suppresses neurons with negative values, (Nwankpa, et al., 2018):

$$f(x) = \max(0, x) \tag{3}$$

The training process adapts all the network parameters so that the input/output relation will be accurate. The hidden layers are functions of the neurons in the previous layers. Thus, we relate the output and the input layers as follows:

$$pred = F\{feature_1, \dots \dots feature_n\} \tag{4}$$

where $feature_n$ represents the n-th input variable. The prediction $pred$ can be compared to the 'true' value ($label$) directly. Therefore, a loss function can be calculated, with the typical formulation for regression purposes, the mean squared error (mse):

$$mse = \frac{1}{n} \sum_{i=1}^{n} (pred_i - label_i)^2 \tag{5}$$

The parameters of the neural network, i.e., the weights and biases, describe the loss function. We aim to minimize the loss function, and thus the set of parameters that best satisfies this condition is defined by the network. The local minimum of the loss function is searched for using stochastic gradient descent.



Another step is the standardization of the data before the training process. This is applied to avoid numerical issues. The mean ($\mu$) and standard deviation ($\sigma$) of the training dataset are used to standardize each feature $x$ separately as follows:

$$x' = \frac{x - \mu}{\sigma} \qquad (6)$$

Thus, the feature variables are scaled to a standard deviation of one and centered around zero. This mainly affects the search for local minima using gradient descent. Introducing features with very different values (and value variations) might result in

a steep gradient descent, leading to a solution that is not optimal. For a deeper look into neural networks, we refer the readers to (Hastie, et al., 2009), (Stanford CS, 2023).

Note that MLP is not necessarily the best algorithm for our research question, but it is not our ultimate goal in this research to find the most appropriate one. Our objective is to demonstrate that ML can be used to map RO data and that it provides comparable (or better) results compared to state-of-the-art methods.

**3.3 Machine learning applied to Bayesian interpolation residuals**

We also develop a method where we combine the BI with ML, named BI&ML. In this case, we train on residuals of the BI. The procedure is as follows:

-    Apply BI to the training dataset. We compute the spherical harmonic coefficients with the same 80% of the data that are used for training in ML.

-    Compute residuals of the BI applied to the training dataset.

-    Train the neural network using as target the BI post-fit residuals (of the training dataset), and as inputs longitude, latitude, and time. The tuning of the hyperparameters is done as explained in section 4.

-    Compute error statistics by comparing BI&ML refractivities with the test dataset.

**4 Analysis**

Here we apply mapping to real and simulated COSMIC-2 data. Our application is an interpolation problem, with the goal of mapping RO data in longitude, latitude and time.

In the first step of the analysis, we compare the post-fit residuals of the BI, ML, and BI&ML approaches based on actual COSMIC-2 RO data. In the second step, we evaluate the performance of BI and BI&ML approaches using the atmospheric analyses of the ECMWF operational products as a nature run. Initially, we compute the (gridded) residuals for simulated

ECMWF refractivities at COSMIC RO locations and then we evaluate the effective horizontal resolution of these two approaches.

To fit the BI and ML models, we use 10 days of COSMIC-2 data and ECMWF analyses. A longer timespan of data does not affect the results of ML, however, BI is more sensitive to the length of data. Indeed, BI is only able to estimate the best





spatial fit to an entire 10-day dataset. The atmospheric state evolves over each 10-day period, and thus BI can only estimate a time-average state with less horizontal structure. Ten days is a compromise to have enough data for properly training the ML models and at the same time to produce BI climatologies with little averaging over time. Unlike BI, in our tests, ML can produce climatologies with a very high temporal resolution.

In the first subsection, we give the results for hyperparameter tuning for ML and BI&ML. In the second subsection, we compare the relative performance of all three approaches described in Section 3 at different heights in the atmosphere. In the third subsection, we analyze the consequences for our ability to resolve spatial and temporal variability in fields that can be measured by RO.

## 4.1 Tuning hyperparameters

One important step when working with neural networks is tuning the hyperparameters, which define the architecture or how the training process is performed. When defining the hyperparameters that determine the architecture of the network, such as the number of layers and number of neurons per layer, one option is to add layers until the error can no longer be reduced. A large number of parameters allows for complexity in mappings, thus, adding more layers can improve the complexity of fitting. However, it can also lead to overfitting. If necessary, regularization methods (such as dropout) can be used to prevent overfitting.

As is customary in ML, we randomly split the nature dataset into two segments: 80% of the data for training, 20% for testing. Since we neither assume different accuracy of training and testing datasets nor teach the network any specific random behavior of the test dataset, we expect the accuracy of the model fitted to the training dataset to be similar for the testing dataset. Therefore, a very good fit in the training dataset does not bias the testing. Indeed, we get similar post-fit residuals for both datasets; this is considered a successful evaluation of the model resulting from the training dataset.

In addition, in one exercise, for each block of 10 days of data, we use the first 8 days for training and the last 2 days for testing. This makes our task a prediction problem and not an interpolation problem. We evaluate the quality of this approach by intentionally overfitting the training data, resulting in overly large errors when applied to the testing data as expected. We can explain this result by the fact that there are different structures in the refractivity field for the testing dataset that have not been seen by the network during the training. This exercise aimed to show the effect of overfitting, which would be an issue for prediction and not for interpolation.

### 4.1.1 Hyperparameter tuning for the ML approach

After tuning, we settled upon a final architecture consisting of 5 hidden layers with the first having 512 neurons and the next four having 128 neurons. It is not unusual for neural networks to have different numbers of neurons in each layer leading to similar error statistics. The input layer consists of three variables, namely longitude, latitude, and time, and the output layer delivers microwave refractivity.





We tuned variables that determine how the training is done, where the choice of learning rate, the batch size and the number of epochs impacted the results the most. The final values that we chose are 0.0001, 100 and 30000, respectively. Note that we designed different neural networks to train refractivities at different altitudes (shown in section 2.1), specifically six heights from 2 km to 20 km. Although we used the same final hyperparameter values for all networks, in some cases a different choice provided similar (but not better) statistics. For instance, at 20 km altitude, for learning rate,

batch size and number of epochs, using 0.0001, 250 and 30000 or 0.001, 250 and 15000, respectively, led to similar results. An example of hyperparameter tuning is shown in Figure 2, where the best results are shown in more distinct colors.

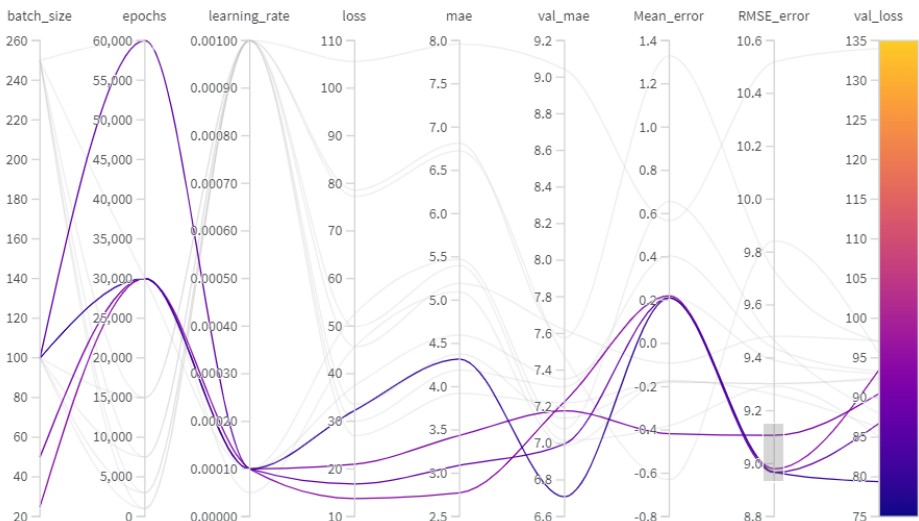

**Figure 2: Hyperparameter tuning in ML, visualized in weights & biases (https://wandb.ai/site). The first three columns contain the values tried during the tuning process for three main hyperparameters (batch size, epochs, and**
**learning rate). The next columns contain statistics that we can use to evaluate the performance of candidate combinations of tuned hyperparameters. The column ("loss") represents the loss function (mean squared error function) of the training dataset. The columns ("mae") and ("val_mae") represent the mean absolute error of the training and validation dataset. The columns ("Mean_error") and ("RMSE_error") represent the mean error and the root mean squared error of the testing dataset. The final column ("val_loss") defines loss function for the**
**validation dataset. The validation loss is the metric chosen to tune the hyperparameters and the colorbar represents its results. Following the curves, we can define the best set of hyperparameters. The validation dataset represents a randomly chosen 10% of the training dataset, utilized to tune the hyperparameters. The statistics of the validation dataset can be easily computed, and therefore, chosen as a metric to select the hyperparameters. The highlighted curves show the best results on the testing dataset in terms of root mean squared error for the different tunings,**
**arbitrarily chosen for this visualization.**

### 4.1.2 Hyperparameter tuning for the BI&ML approach

Once more, we trained different neural networks for six refractivity altitudes from 2 km to 20 km. The architecture of the MLP is the same as that shown in section 4.1.1, consisting of 5 hidden layers with the first having 512 neurons and the next four having 128 neurons. The input layer has three variables, namely longitude, latitude, and time, and the output layer

delivers BI residuals of microwave refractivity. The learning rate is again 0.0001 for all networks. The tuned batch size and



number of epochs are 25 and 2000, respectively, for the networks of 2 km, 3 km and 5 km altitudes, 50 and 1000 for the network of 8 km altitude, and 100 and 4000 for the networks of 15 km and 20 km altitude. Again, other possible hyperparameter choices could result in similar (but not better) results; for example, we could choose for the dataset at 5 km altitude a learning rate of 0.001, batch size of 250 and number of epochs of 1000. Different tunings with similar results were especially encountered when training data at high altitudes. One reason is that the target value variations become very small and it is possible to learn them with different values for learning rate, number of batch size and/or number of epochs. Training BI residuals, instead of total refractivity values, leads to a faster training process. One explanation is that the network can learn more quickly when the targets have smaller variations.

## 4.2 Performance evaluation for real RO data

Using the BI and ML models obtained from the training dataset, we mapped the observations of the test dataset, which is 20% of the data, and then we computed the residuals for the six chosen heights. Figure 3 displays the residuals for the three methods (BI, ML, and BI&ML) and Table 1 summarizes the statistics in terms of standard deviation (std) and mean relative error (MRE). The MRE represents the mean of the residuals scaled by the true values of the refractivity.

- As expected, the residuals are higher at lower altitudes for each method. This is logical since the refractivity values are higher and more variable at lower altitudes (see Figure 3), and the Abel inversion results in larger errors at lower altitudes. Therefore, the noise of the observations is also higher at lower altitudes.

- The spatial distribution of the larger residuals is different for the different heights. For instance, at the lower heights, we can find most of the high residuals of BI in the tropics, while at 15 km they are located in the mid latitudes. This is the case since the mapping methods miss the high jumps of the refractivity values due to approximation of RO data and low resolution of RO data. At 2 km and 3 km these jumps occur mainly in the tropics, related to higher water vapor, and in the mountainous areas, where the distribution of water vapor can highly vary on the different sides. At 15 km, the mapping functions fail to approximate the large refractivity jumps between the troposphere (in the tropics) and the stratosphere (in mid-latitudes).

- We notice that when we apply ML, the refractivity jumps at 15 km from the tropics to the mid-latitudes are well captured.

- From Figure 3, it might be difficult to understand the benefit of applying ML to RO residuals after applying BI. However, from the statistics in Table 1, we notice an improvement of about 5-10% compared to ML-only. One advantage of applying ML to RO residuals is in terms of interpretation of the dataset used to fit the ML model. For instance, at 15 km we expect ML to learn from the distinct pattern of BI residuals (where the large residuals occur at mid-latitudes), and to further improve these results. In addition, as mentioned in Section 4.1.2, ML applied to RO residuals is much more efficient in terms of time to train the ML models. In our experiments, the time to train the neural network is proportional to the number of epochs and the batch size. For example, in case of two different





models, if we use the same batch size, a two times larger number of epochs increases the training time twice. Similarly, if we consider the same number of epochs a two times smaller batch size increases the training time twice. Other hyperparameters can also affect the training time such as the type of optimizer, the number of neurons and the learning rate (smaller learning rate requires larger number of epochs); however, these parameters are the same for the different neural networks that we use.


The overall best configuration, which we will focus on section 4.3, is the ML approach applied to the residuals of BI.





**Figure 3:** Residuals for the test dataset of the refractivity interpolated with BI, ML and BI&ML for the six pre-defined isohypsic surfaces. The units are N-units (refractivity units), the x-axis is east longitude, and the y-axis is northward latitude. The plots (a), (d), (g), (j), (m), (p) display BI residuals at 2, 3, 5, 8, 15 and 20 km, respectively. The plots (b), (e), (h), (k), (n), (q) display ML residuals at 2, 3, 5, 8, 15 and 20 km, respectively. The plots (c), (f), (i), (l), (o), (r) display BI&ML residuals at 2, 3, 5, 8, 15 and 20 km, respectively.





| N unit/% | 2 km | 3 km | 5 km | 8 km | 15 km | 20 km |
|---|---|---|---|---|---|---|
| **Std BI** | 10.95 | 9.11 | 5.28 | 1.59 | 0.65 | 0.29 |
| **Std ML** | 8.97 | 6.98 | 3.82 | 1.16 | 0.28 | **0.22** |
| **Std BI&ML** | **8.71** | **6.65** | **3.57** | **1.09** | **0.26** | **0.22** |
| **MRE BI** | 3.34 | 3.19 | 2.33 | 1.03 | 0.92 | 0.95 |
| **MRE ML** | 2.62 | 2.34 | 1.63 | 0.7 | 0.42 | **0.62** |
| **MRE BI&ML** | **2.57** | **2.23** | **1.50** | **0.66** | **0.39** | **0.62** |

**Table 1: Statistics of BI, ML and BI&ML mapping methods, at six predefined heights, for Winter 2020.**

Figure 4 displays the increment of BI&ML over the ML-only and BI-only approaches, at 3 km altitude. The increment of BI&ML over BI-only has large values mainly in the geolocations, where we can visualize large jumps of refractivity within few degrees (latitude and/or longitude). The increment of BI&ML over ML-only has a more random distribution of large values. These values happen mainly in locations where the residuals of ML and/or BI&ML (see Figure 3) have a larger

density. By training on BI residuals, we anticipated better results since a part of the refractivity behavior is already removed; however, since the spatial resolution of RO data is not very high, BI does not always capture the spatial refractivity changes very well (especially when sudden refractivity changes happen). This may lead to higher relative changes between the residuals compared to the total values, thus, the trained values in the ML model have larger variations. In addition, since BI is a screen over the 10 days, the mapped refractivity, for similar locations, can be more accurate for some epochs of the 10-

day timespan. Since we do not label the noise of each input feature differently when we train the ML models, ML will consider all residuals as having the same weight, which will impact its mapping accuracy. However, from the statistics in Table 1, we can see that ML applied to BI residuals results in better generalization for the entire timespan and surface. Indeed, it removes most of the complex atmospheric dynamics and simplifies the entire variation of the target variables.

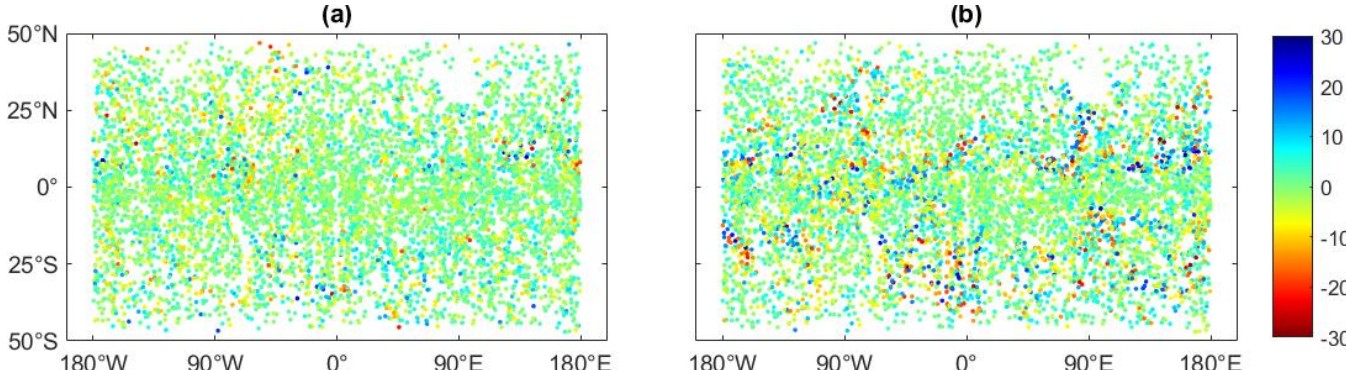

**Figure 4: Difference between BI&ML and ML-only, plot (a), and difference between BI&ML and BI-only, plot (b). The units are N-units, refractivity units, the x-axis is east longitude, and the y-axis is northward latitude.**



We expect ML-only to be the best approach when we deal with specific atmospheric structures happening at a specific day of the 10-day timespan. For instance, in case of atmospheric rivers, BI would fail to capture these structures, since it will find the best fit of the 10-day dataset. Therefore, it will result in large variations of the residuals, making them more complex to

train on than total refractivities.

### 4.2.1 Results in different seasons at 2 km

We also validated the results obtained for Winter 2020 with the other seasons of the same year. Table 2 summarizes the results of the three mapping methods for each season at 2 km altitude. At 2 km altitude, RO observations are noisier compared to the other altitudes, because of larger absolute values and larger errors resulting from the Abel inversion. In

addition, their uncertainty and variation are higher, since it is at this height that the largest part of water vapor is located. For all the seasons, the best performance is achieved for the combined solution, resulting in a further improvement compared to BI-only and ML-only. Note that again each of the neural networks was tuned separately to achieve the best possible solution.

| N unit/% | Winter | Spring | Summer | Autumn |
|---|---|---|---|---|
| **Std BI** | 10.95 | 10.54 | 10.63 | 10.90 |
| **Std ML** | 8.97 | 9.01 | 8.59 | 8.82 |
| **Std BI&ML** | **8.71** | **8.39** | **8.31** | **8.70** |
| **MRE BI** | 3.34 | 3.21 | 3.21 | 3.31 |
| **MRE ML** | 2.62 | 2.67 | 2.55 | 2.63 |
| **MRE BI&ML** | **2.57** | **2.48** | **2.45** | **2.59** |

**Table 2 Statistics of BI, ML and BI&ML mapping methods, for the four different seasons, at 2 km height.**

### 4.3 Post-fit residuals of BI, ML and BI&ML mapping techniques applied to ECMWF

We compare the post-fit residuals of the BI approach, the ML approach, and the combined BI&ML by applying each to the 10-day nature run of ECMWF forecast products.

We perform a similar evaluation for ECMWF as we did for the COSMIC-2 data, where we split the data in training and test samples and apply the three methods. We obtain, in terms of standard deviation and mean relative error, 12.4 N-units and 3.8% for BI, 11.1 N-units and 3.3% for ML and, 10.7 N-units and 3.1% for BI&ML. We confirm that mapping ECMWF

(forecast) refractivities (interpolated at COSMIC-2 geolocations) results in a performance similar to the mapping of COSMIC-2 refractivities.

### 4.3.1 Structural improvement: Spatial and temporal resolution compared to ECMWF maps

We mapped the ECMWF-based simulated refractivities with BI and BI&ML to the same locations as the ECMWF grid points. An example of the maps (at 2 km height) is displayed in Figure 5, where the original ECMWF N-field for one epoch





is displayed as well. We can see the very high resolution of the original ECMWF map, compared to the interpolated ones. The BI-mapped field is the screen over the 10-day dataset, while that of the BI&ML represents the interpolation at only one epoch. We produced maps with a resolution of 3 hours. The ML-based maps are produced with much higher temporal resolution compared to those with BI (a resolution of 3 days reported in previous works, (Leroy, et al., 2021)). In addition, Figure 6 displays the difference between the original ECMWF and the mapped refractivity fields. We notice that there are

several areas where the high differences between ECMWF and BI-based map are further smoothed by applying ML to the BI residuals.

Figure 6 displays only one map of the total of 80 maps we produced for the 10-day period. To compare the results for each epoch, Figure 7 displays the statistics for every map as a function of time. There is an improvement in the interval [0.5; 2] N-units and [0.5; 1] % in terms of standard deviation and the mean relative error, respectively.

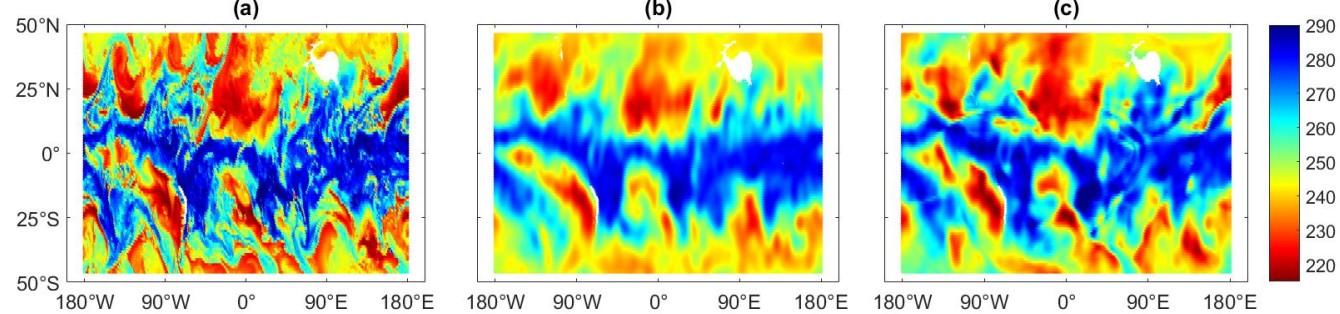

**Figure 5: ECMWF refractivity grid, plot (a), BI mapped refractivity grid, plot (b), and BI&ML mapped refractivity grid, plot (c), for 2nd January 2020 at 03:00. The units are N-units, refractivity units, the x-axis is east longitude, and the y-axis is northward latitude.**

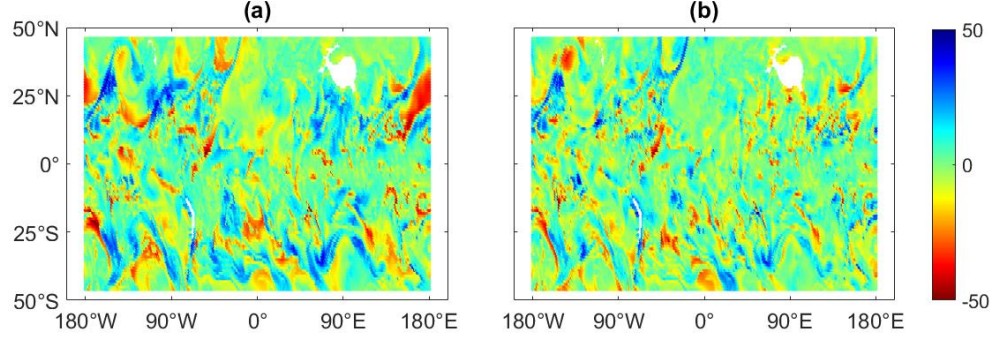

**Figure 6: Difference between ECMWF refractivity field and the refractivity fields mapped with BI, for 2nd January 2020 at 03:00, plot (a). Difference between ECMWF refractivity field and the refractivity fields mapped BI&ML, for 2nd January 2020 at 03:00, plot (b). The units are N-units, refractivity units, the x-axis is east longitude, and the y-axis is northward latitude.**




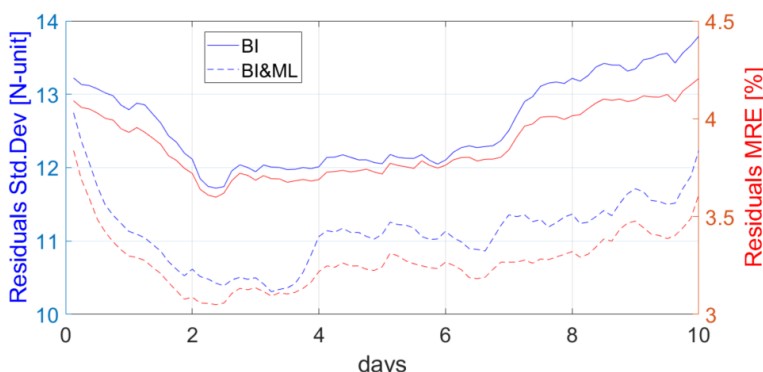

**Figure 7: Standard deviation (left y-axis) and mean relative error (right y-axis) of the difference between the ECMWF refractivity field and those mapped with BI and BI& ML for all epochs.**

**4.3.2 Effective horizontal resolution**

After fitting the BI and BI&ML models, we can produce refractivity maps with a very high spatial resolution. However, the interpolated resolution is not the actual resolution that the methods can capture for the spatial behavior of the refractivity. To

evaluate what level of information BI and BI&ML can produce in terms of horizontal resolution, we use the original and mapped fields to compute spherical harmonic spectral coefficients up to a very high order (such as 120). We visualize the power (and variance of the fit dataset compared to those of ECMWF) as a function of degree (or horizontal resolution).

Each of the datasets can be expressed as a double sum, (Muir & Tkalcic, 2015):

$$\psi(\theta, \lambda) = \sum_{l=0}^{l_{max}} \sum_{m=-l}^{m=l} Y_m^l(\theta, \lambda) c_{l,m} \tag{7}$$

where $Y_m^l(\theta, \lambda) = N_{l,m} P_{l,m}(sin\,\theta) e^{im\lambda}$ are functions of the normalized associated Legendre polynomials. $c_{l,m}$ are spherical harmonics spectral coefficients, where $l$ and $m$ define the degree and order. We can compute the spherical harmonics spectral coefficients by inverting Eq. (7):

$$c_{l,m} = \sum_{j=1}^{n_g} \left\{ \frac{1}{2\pi} \int_0^{2\pi} \psi(\theta_j, \lambda) e^{-im\lambda} d\lambda \right\} P_{l,m}(sin\,\theta_j)\, g_j \tag{8}$$

which is computed over a defined Gaussian grid for $j = 1 : n_g$ over the latitude $\theta$, with Gaussian weights $g_j$. We can

compute the power at each degree as (Muir & Tkalcic, 2015):

$$P(l) = \sum_{m=0}^{l} |c_{l,m}|^2 \tag{9}$$

In addition, we can compute the normalized power $P\_norm(l) = P(l)/Hor\_res(l)$, where the horizontal resolution is given as a function of the Earth's radius $R_E$, $Hor\_res(l) = R_E\sqrt{4\pi}/(l+1)$.

We also evaluate the explained variance, defined as:

$$Explained\ variance(l) = 1 - \frac{Var(l:Fit - ECMWF)}{Var(l:ECMWF)} \tag{10}$$





Considering $d_m = 1$ for $m = 0$ and $d_m = 2$ otherwise, the two variances can be computed:

$$Var(l; Fit - ECMWF) = \sum_{m=0}^{l} d_m |c_{l,m}(BI \text{ or } BI\&ML) - c_{l,m}(ECMWF)|^2 \tag{11}$$

$$Var(l; ECMWF) = \sum_{m=0}^{l} d_m |c_{l,m}(ECMWF)|^2 \tag{12}$$

To avoid non-orthogonality of spherical harmonics on non-global grids, we must consider observations covering the entire

sphere. We randomly select values from the 10 days of the ECMWF forecast data grid, at latitudes that are not covered by

the observations, i.e., outside the [-46°, 46°] latitude interval. The only condition we apply is to have the same spatial density

as that in the interval [-46°, 46°] latitude. An example of the refractivity field (at 2 km), for 10 days in Winter 2020, is shown

in Figure 8. This refractivity field is used to fit the BI and BI&ML models and therefore map the refractivity on the original

ECMWF grid.

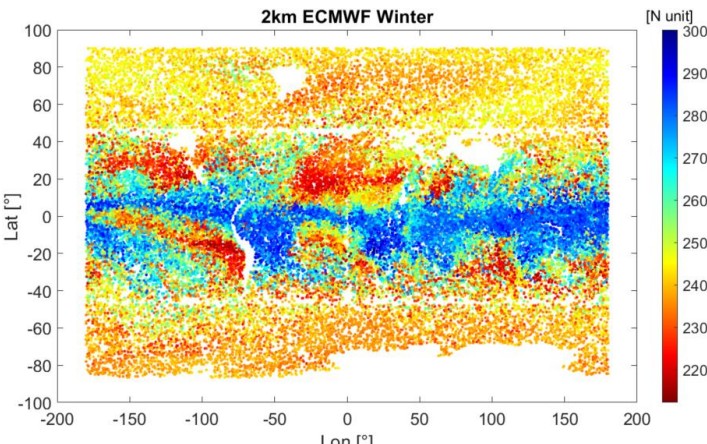


**Figure 8: ECMWF forecast refractivity field for 10 days in Winter 2020, at 2 km isohypsic surface. To compute spherical harmonics, we simulated refractivity at all latitudes. Between 46°S and 46°N the refractivity is interpolated at the same times and locations as for the COSMIC-2 field. Below 46°S and above 46° N, we randomly chose refractivity values from ECMWF data, with the criteria to keep the same density as that of COSMIC-2 between 46°S**

**and 46°N latitude.**

After fitting the BI and BI&ML models, we produce gridded N fields (similar to Figure 5), which are used to compute the

spherical harmonics spectral coefficients (Eq. (8)). Figure 9 and Figure 10 display the normalized power and the explained

variance for two of the evaluated heights, 2 km and 8 km. The power and explained variance displayed here are the average

over the entire dataset of 80 maps in 10 days. From the power plots, we notice that the BI&ML curve (red dots) follows the

ECMWF curve at a higher spherical harmonics degree (and thus horizontal resolution). We can also see that after 40 degrees

the power of BI is zero. This is expected since the degree we chose for the BI was 40 as well.

We evaluate the explained variance at a value of 0.5, which represents the value where the captured power from the mapping

methods is half of the original ECMWF power. We use this value as the metric to define the horizontal resolution of each

method. From the explained variance plots in Figure 9 and Figure 10, we can see that BI can capture horizontal structures up





to degree 8 (~2500 km) at 2 km height and up to degree 4 (~4500 km) at 8 km height. BI&ML can produce maps with a horizontal resolution of degree 14 (~1500 km) at 2 km height and of degree 16 (~1250 km) at 8 km height. These results represent the average over the entire 80 maps. For both heights and the majority of the produced maps, applying ML to BI residuals improves the spatial resolution compared to the BI-only approach.

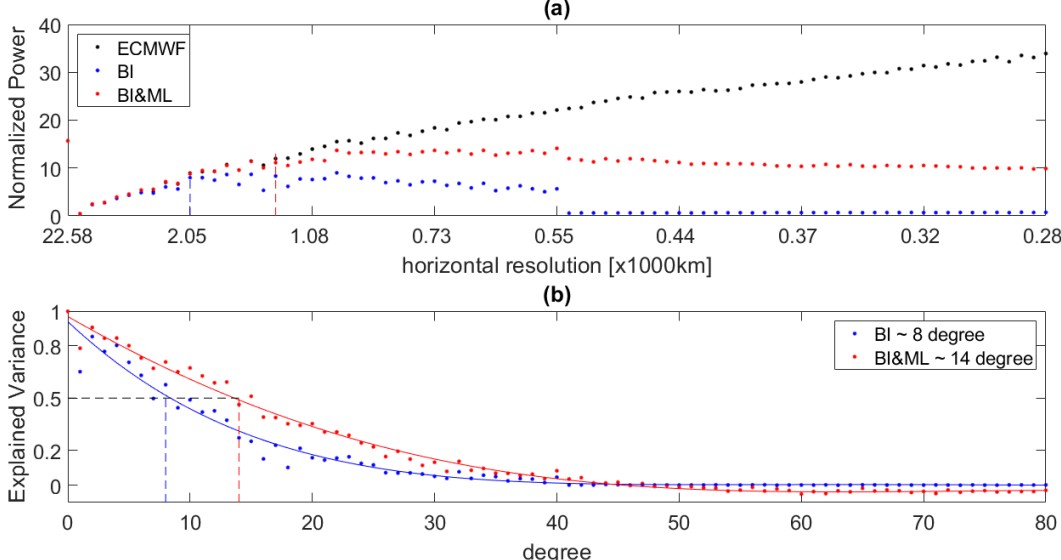

**Figure 9: Plot (a): average (over time) normalized power for ECMWF, BI and BI&ML, at 2 km height. Plot (b): average (over time) explained variance for BI and BI&ML, at 2 km height.**

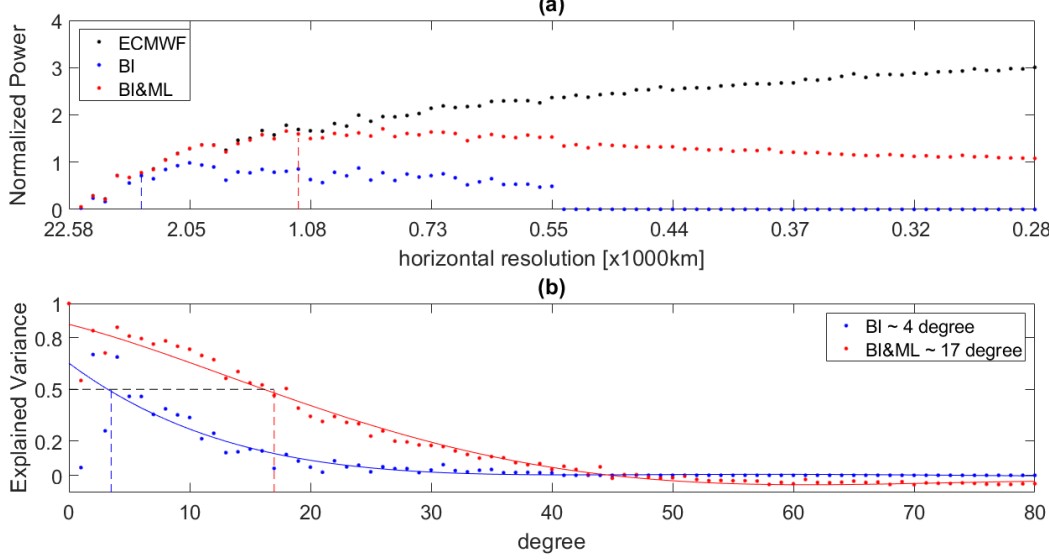

**Figure 10: Plot (a): average (over time) normalized power for ECMWF, BI and BI&ML at 8 km height. Plot (b): average (over time) explained variance for BI and BI&ML at 8 km height.**



## 5 Discussion and conclusions

In this work, we investigated ML as an alternative approach to mapping GNSS RO observations and compared it to BI, an approach studied for many years to map GNSS RO data. In addition, we developed a combined solution, where we map residuals of BI using ML, referred to as BI&ML.

Starting with 10 days of COSMIC-2 GNSS RO profiles in boreal winter 2020, we mapped microwave refractivity at six predefined heights: (a) 2 km where we notice small structures related to boundary layer clouds and water vapor; (b) 3 km where there is still a large amount of water vapor; (c) 5 km related to synoptic disturbances, pressure fields, storms and precipitation; (d) 8 km similar to 5 km but with smaller refractivity; (e) 15 km where the eddy mixing in the stratospheric "middle-world" occurs; and (f) 20 km related to larger atmospheric structures from planetary waves in the lower stratosphere.

We used 80% of the COSMIC-2 data to train/fit the BI, ML and BI&ML models and evaluated the performance on the remaining 20%. The ML-only solution results in better performance than the BI-only solution. Applying ML to the residuals of BI results in the best performance and a larger improvement compared to the state-of-the-art BI method. The posterior uncertainties for BI&ML are 8.7, 6.6, 3.6, 1.1, 0.3 and 0.2 N-unit, and the mean relative errors for BI&ML are 2.57, 2.23, 1.5, 0.66, 0.36 and 0.62 %, respectively, for the six altitudes. The reduction of residuals for the ML-only and ML&BI at 15 km compared to BI-only are clearly visible (Figure 6), where the refractivity values change significantly between the tropics (in the troposphere layer) and the mid-latitudes (in the stratosphere layer). In addition, we fit the BI, ML and BI&ML mapping approaches to 10 days of COMSIC-2 data for boreal spring, summer and autumn 2020. We performed this evaluation at 2 km iso-height, and we confirmed that the results obtained for the winter scenario apply to the other seasons as well.

We used NWP forecasts from ECMWF with a 0.5° latitude/longitude resolution to interpolate ECMWF refractivities to the geolocations and times of the COSMIC-2 data, for 10 days for boreal winter 2020. We applied BI-only and BI&ML to map the simulated data at the ECMWF grid points, which we used as a nature run and then performed a closed-loop validation. The map produced by BI&ML shows smaller residuals with respect to the nature ECMWF map than the BI-only map. In addition, the temporal resolution of the BI&ML maps is much higher than that of the BI-only maps. We produced BI&ML-based maps every 3 hours (higher resolution is also possible), while BI-only needs at least 3 days of observations to produce a map. We investigated the spatial resolution of each method by spherical harmonic expansion, comparing the spectral coefficients of the BI-only and the BI&ML maps to the spectral coefficients of the nature ECMWF maps. After evaluating the explained variance of the coefficients, we concluded that BI-only can model refractivity variations up to spherical harmonic degree 8 and BI&ML up to spherical harmonic degree 14 at 2 km iso-height. At 8 km height, the improvement is more notable, with BI-only resolving only up to spherical harmonic degree 4 while BI&ML resolves up to spherical harmonic degree 17.



We set out to investigate whether ML can offer an alternative to existing methods to map GNSS RO data, providing a so-called level 3 product. Existing methods include BI, which fits data using spatial basis functions without over-fitting data, and sampling-error-removal methods, in which synoptic variability noise is estimated by subsampling the forecasts of a

numerical weather prediction system to the times and locations of RO soundings, computing the sampling error, and subtracting that sampling error from binned RO data. We compared ML methods to BI and found improved performance, and then we compared a combined BI&ML method and found a very substantial improvement of the performance over BI. We are unable to compare to sampling-error-removal approaches because they support no error estimation. All indications point toward the combined BI&ML approach as the best method for producing level 3 climatologies of RO data in the

future, with strong performance even at time-scales of 3 hours at heights ranging from the planetary boundary layer up to the lower stratosphere and for all seasons.

*Code and data availability.* The code associated with this study is available from the corresponding author on reasonable request. The datasets generated during and/or analyzed during the current study are available from the corresponding author on reasonable request. The COSMIC-2/FORMOSAT-7 data is freely available from UCAR. The ECMWF data are a product

of the European Centre for Medium-Range Weather Forecasts (ECMWF) (© ECMWF).

*Author contributions.* The work presented in this paper was carried out in collaboration between all authors. ES did the preprocessing of the data, implemented the experiments, and performed data analysis. SL implemented the Bayesian Interpolation scripts, provided advice on data analysis and contributed to the interpretation of the results. KC provided advice on data analysis and contributed to the interpretation of the results. AG, GM, MR contributed to the interpretation of

the results. LC and BS provided the ECMWF data and helped with the interpretation of the results. ES and SL prepared a draft of the paper, and all authors edited the manuscript.

*Competing interests.* The authors declare that they have no conflict of interest

*Acknowledgements.* The authors would like to thank the Swiss National Science Foundation (SNSF) for financing this work (project number 200021E-168952). S. Leroy was supported by the Decadal Survey Incubator Science program of the U.S.

National Aeronautics and Space Administration, grant 80NSSC22K1003.

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
