# Peer review of "GNSS Radio Occultation Climatologies mapped by Machine Learning and Bayesian Interpolation"

_Atmospheric Measurement Techniques, 2023_

## Referee Comment (RC2)

The paper proposes an innovative method that combines Bayesian interpolation (BI) and basic Multi-Layer Perception (MLP) to map refractivity from Global Navigation Satellite Systems (GNSS) radio occultation (RO) data, i.e., COSMIC-2. While the study suggests that the BI&ML model outperforms individual BI and MLP models, there are some concerns regarding the presented conclusions.

Review Comments:

1. Training Consistency: The paper suggests that BI&ML outperforms MLP alone, but upon examining tables 1 and 2, the difference does not appear significant. It is common knowledge that the performance of the MLP model is tied to its initial state and training quality. To address this concern, it is recommended that the author trains multiple models for all methods to provide a clearer understanding of uncertainty associated with each model.

2. Hyperparameter Tuning: Figure 2 illustrates hyperparameter tuning on wandb, which is commendable. However, it would be beneficial to explore more critical hyperparameters, such as the optimization method (e.g., Adam, AdamW, RMSprop), number of layers, and weight decay. These parameters are likely to have a more substantial impact than batch size, epochs, and learning rate.

3. Data Preprocessing: The paper does not explicitly mention any preprocessing steps for COSMIC-2 data. It would be insightful to provide details on any preprocessing carried out, as this could significantly influence the model's performance.

4. Data Splitting Strategy: Randomly splitting data into train, valid, and test sets might not be optimal, as these sets could be interrelated (so called 'data leaking'). The suggestion is to have an independent validation set and test set for a more robust evaluation of the proposed models.

5. Code and Data Availability: To enhance the reproducibility and validation of the research, it is recommended that the authors provide complete code and data. This would enable other researchers to replicate the experiments more easily and validate the results effectively.

---

## Referee Comment (RC3)

Review Comments to the manuscript on "**GNSS Radio Occultation Climatologies mapped by Machine Learning and Bayesian Interpolation**" submitted to Journal of AMT.

Overall Comments:

The work carried out by the authors in the submitted manuscript is of great significance for regularizing the sparsely available GNSS radio occultation (RO) data on a global grid. Such a data may be gainfully used for morphological construction of RO climatology. The technique of machine learning (ML) developed in the work is an advancement over Bayesian Interpolation (BI) – both as standalone model as well as when combined with BI. It is also a timely application of the current global use of AI/ML approaches, especially, with a copious amount of GNSS RO data being available from past and existing satellite missions. The manuscript is clear in its objectives, cogent and systematic in its presentation and well-written in lucid language. Though the authors have liked to restrict their objective to showcase the benefit of ML – as standalone and as combined model with BI, it would have been thorough if some of the advanced ML models such as those based on decision trees viz. random forest, XGBoost etc or using several regressors like stacking regressor were also compared in terms of their statistical metrics. However, this is just a suggestion for future work. **I strongly recommend for the manuscript to be accepted for publication in the journal of AMT after incorporation of corrections suggested as minor comments below.**

Other minor comments:

1. Line 107: wetPf2 is not the refractivity rather it is the name assigned to a set of retrieved state parameters using 1dVAR method. It is better to state "analyzed refractivity sourced from wetPf2 files from the data portal ….".
2. Authors to precisely refer in texts (Lines 115-125) to each sub-figures using the assigned alphabets in figure 1. What is the grid size along the zonal and meridional direction chosen for each sub-figures of figure 1.
3. Lines 145-150: Authors to mention whether they have accounted for the difference between geopotential fields for interpolated refractivity from ECMWF model forecasts and geometric altitude above mean sea level for COSMIC-2 refractivity before comparing?
4. In line 145, what is the reason for not using any prior forecast fields such as 3 hours, 6 hours? Is it availability or any other justifiable reason?
5. In line 464: correct the combined model name to BI&ML.

---

## Author Comment (AC1)

**Associate editor comment:**

**If the authors submit a final version of the manuscript, then it will be necessary to include some evidence (a figure or a reference) showing that the current spatial and temporal density of RO leads to a need for better coverage when approaching specific goals.**

Dear Associate Editor, thank you very much for the comment.

We use a scenario of an Atmospheric River (AR) to show the importance of higher temporal and spatial RO density to detect such a structure. We provide an example of how we need improved spatial and temporal density of RO measurements for the case of an atmospheric river, and we have added a figure (Figure 11) to the manuscript to help clarify this.

We have displayed such experiments in *Shehaj 2023, Space Geodetic Techniques for Retrieval of High-Resolution Atmospheric Water Vapor Fields, PhD thesis, ETH Zurich, No. 29245, Zurich, Switzerland.* The following plots summarize the results in the thesis. The assessment is performed for a height of 2 km.

We identified an AR scenario on the West Coast of the US (on the website of US National Weather Service, NOAA, *«National Weather Service,» [Online]. Available: https://www.weather.gov/mtr/Atmospheri-cRiver_10_24-25_2021. [Accessed 15 February 2023]*), during 24th and 25th of October 2021. The AR (blue stream) hits the US coast from the Pacific Ocean, in the right plot of Figure 1. This AR was visualized using ECMWF hourly forecasts.

[Figure]

Figure 1: River scenario in ECMWF data, refractivity at 2 km height, 24 October 2021 at 19:00 (similar to *Shehaj 2023*), (not in manuscript).

To evaluate our method to detect such structures, we utilize RO observations for a simulated constellation of 60 satellites. In total, we obtain 743499 occultations globally for 4 days (23-26th of October), while in the study region [10°N to 60°N latitudes and 90°W to 160°W longitudes] the number of occultations is 53995. Similar tests were done for 48, 24 and 12 satellites constellations. Figure 2 displays the RO refractivity at 2 km altitude.

[Figure]

Figure 2: Training datasets for a simulated constellation of 60 satellites (similar to *Shehaj 2023*), (not in manuscript).

Figure 3 displays the ECMWF and the ML mapped refractivity for the 60 satellites constellation. Similar experiments were performed for the 12, 24, and 48 satellites shown in *Shehaj 2023*. We showed that for this assessment a 48-satellite constellation can detect the AR structure at 2 km quite well.

[Figure]

Figure 3: ECMWF reference refractivity field and the refractivity field obtained using ML, 24 Oct 2021 at 16:00, (not in manuscript).

We point out that in *Shehaj 2023*, similar experiments were performed for this AR scenario using observations of the COSMIC-2 constellation; the number of RO observations is much smaller compared to the simulated example shown here. After applying the ML model to grid the COSMIC-2 refractivity (at 2 km height) we cannot properly observe the spatial and temporal evolution of the AR structure.

In the paper, in the discussion and conclusions section, we have added the following section:
'When approaching specific atmospheric structures, the current spatial and temporal density of RO observations leads to a need for better coverage. (Shehaj, 2023) shows an example of using RO observations to detect atmospheric rivers (AR). ARs are long and narrow bands in the atmosphere that transport water vapor in regions beyond the tropics. Plot (a) of Figure 11 shows an AR that occurred in October 2021, visualized as a blue stream using ECMWF refractivity at 2 km altitude. In plot (b) of Figure 11, ML was applied to simulated RO using ECMWF 12-hour forecast, assuming a 60-satellite LEO constellation tracking four GNSS constellations to generate a 2 km refractivity field. For the example in Figure 11, we can see that with the ML-mapped field, we can resolve the AR structure. The ML field also depicts the dry patch located in the tropics. We also notice that there are structures more difficult to resolve, such as the cyclone close to British Columbia, as well as the high refractivity patch close to Hawaii.

[Figure]

**Figure 11: Refractivity at 2 km altitude example for 24 Oct 2021 at 16:00, similar to (Shehaj, 2023). Plot (a): ECMWF reference field. Plot (b): the field mapped using ML (for the 60 satellites constellation).**

We point out that in (Shehaj, 2023) similar experiments were performed for this AR scenario using observations of the COSMIC-2 constellation; the number of RO observations is much smaller compared to the simulated example shown here. After applying the ML model to grid the COSMIC-2 refractivity (at 2 km height), we cannot properly observe the spatial and temporal evolution of the AR structure.

The example in Figure 11 shows the need for higher spatial and temporal density of RO observations and the benefit of using ML as a method to further enhance the resolution of the observations. In future studies, we will further explore the feasibility of GNSS RO for detection and monitoring of ARs.'

---

## Author Comment (AC2)

**Reviewer #1:**

**General comment:**

**This paper is concerned with constructing GNSS RO-based climatologies by machine learning (ML) method, and proposes three kinds of approaches: Bayesian Interpolation (BI), a feed-forward neural network (Multilayer Perceptrons, MLPs), and the combination of BI and ML (BI &ML) where the ML is applied to BI residuals. Applications of these methods to real and simulated COSMIC-2 RO data indicate that, the maps of refractivity produced by the MLPs better match the true maps than those by BI, and BI & ML yields the best GNSS RO refractivity maps. The methods are novel and the results exhibit the potential for producing GNSS RO climatologies.**

Dear reviewer, thank you very much for taking the time to review our work, we appreciate your input. Below, you can find our answers to all your comments.

**specific comments:**

**"BI & ML" is about learning on residuals, which is a key strategy suggested by the authors. I think it is better to provide some reviews for learning on residuals in the introduction.**

We cite the following papers that focus on learning on residuals:

[1] Wang, JX., Wu, JL., Xiao, H. "Physics-informed machine learning approach for reconstructing Reynolds stress modeling discrepancies based on DNS data" Phys Rev Fluids, 2 (3) (2017), Article 034603

[2] Gou, J.; Rösch, C.; Shehaj, E.; Chen, K.; Kiani Shahvandi, M.; Soja, B.; Rothacher, M. "Modeling the Differences between Ultra-Rapid and Final Orbit Products of GPS Satellites Using Machine-Learning Approaches." *Remote Sens.* **2023**, *15*, 5585. https://doi.org/10.3390/rs15235585

[3] Kiani Shahvandi, M., Dill, R., Dobslaw, H., Kehm, A., Bloßfeld, M., Schartner, M., et al. (2023). "Geophysically informed machine learning for improving rapid estimation and short-term prediction of Earth orientation parameters." Journal of Geophysical Research: Solid Earth, 128, e2023JB026720. https://doi.org/10.1029/2023JB026720.

In the paper, at the end of the introduction, we add the following paragraph: 'Several studies have also applied ML to model residuals of observations, computed as a difference between a model not based on ML and target observations. The most typical cases originate from using physical models for prediction and training an ML model to predict the residual part. For example, (Wang, et al. 2017) showed that ML could be used to model the difference between a superior model which is computationally expensive and a simple model, to predict the component of the total stress tensor in a fluid. Similarly, (Gou, et al. 2023) applied several ML and deep learning (DL) algorithms to model the differences between GNSS final orbit products and ultra-rapid orbit products. Therefore, their ML model could help overcome the limitations of simplified physics-based orbit propagators by training on residuals. (Kiani Shahvandi, et al. 2023) used a method based on NNs named ResLearner to calibrate the rapid Earth Orientation Parameters (EOPs) with respect to the final EOPs in a residual manner. In this work, we also propose a loosely coupled combination of ML and BI, in which we first apply BI to the observations, and then we train the ML model on the BI residuals. '

**technical corrections:**

**For the convenience of readers, Figure 3, Figure 4, Table 1 and Table 2 had better be indicated for COSMIC -2 RO data.**

Thank you for pointing this out. This will be properly indicated in the final version of the manuscript.

---

## Author Comment (AC3)

**Reviewer #2:**

**The paper proposes an innovative method that combines Bayesian interpolation (BI) and basic MultiLayer Perception (MLP) to map refractivity from Global Navigation Satellite Systems (GNSS) radio occultation (RO) data, i.e., COSMIC-2. While the study suggests that the BI&ML model outperforms individual BI and MLP models, there are some concerns regarding the presented conclusions.**

Dear reviewer, thank you very much for taking the time to review our work, we appreciate your comments. Below, you can find our answers to all your points in an extended version.

**Review Comments:**

**1. Training Consistency: The paper suggests that BI&ML outperforms MLP alone, but upon examining tables 1 and 2, the difference does not appear significant. It is common knowledge that the performance of the MLP model is tied to its initial state and training quality. To address this concern, it is recommended that the author trains multiple models for all methods to provide a clearer understanding of uncertainty associated with each model.**

Thank you for your comment. To address this comment, we performed further experiments where we trained several models.

In Table 1, we summarize the results on an ensemble of 10 neural networks for ML and BI&ML. We display these results for a network of 5 layers and for a network of 8 layers (related to your second comment). Similarly to the results in the paper, there is a smaller standard deviation for the BI&ML compared to ML, for a network with 5 layers. For a network with 8 layers, the improvement is smaller, however the uncertainty of the standard deviation is also improved.

Note that Table 1 has not been included in the paper, however, we have added the following paragraph: 'Note that the results displayed in this section (Table 1 and Table 2), are a result of one single trained network, and not of an ensemble of networks. To further validate our results, we performed additional experiments for the refractivity at 2 km iso-height, where we trained multiple (10) models for ML and BI&ML. For our architecture, similar results were achieved on the results of the ensemble of the models, with ~0.2 N-unit worse standard deviation. '

*Table 1: Statistics (in N-unit) over 10 trained models on ML and BI&ML models, (not in manuscript).*

|  | 5 Layers | | | | 8 Layers | | | |
|---|---|---|---|---|---|---|---|---|
|  | Av. STD | Std. St.dev | Av. Bias | Std. Bias | Av. STD | Std. St.dev | Av. Bias | Std. Bias |
| **ML** | 9.16 | 0.08 | 0.19 | 0.31 | 8.90 | 0.13 | 0.41 | 0.17 |
| **BI&ML** | **8.96** | **0.08** | **0.17** | **0.45** | **8.85** | **0.09** | **0.28** | **0.35** |

**2. Hyperparameter Tuning: Figure 2 illustrates hyperparameter tuning on wandb, which is commendable. However, it would be beneficial to explore more critical hyperparameters, such as the optimization method (e.g., Adam, AdamW, RMSprop), number of layers, and weight decay. These parameters are likely to have a more substantial impact than batch size, epochs, and learning rate.**

Thank you for this input. Following your comment we have trained also the other hyperparameters that you suggested. In Table 2 we have summarized all the results. We point out that all the statistics are computed over 10 different models for each tunned parameter.

Firstly, the main hyperparameter that further impacts the results is the number of layers. Smaller number of layers results in worse results, especially in terms of standard deviation and larger number of layers further improves the results (again, mainly in terms of standard deviation). We also point out that when we trained networks with 7 (or 8) layers and 9 (or 10) layers the time to train the model was about 1.4 and 1.5 time longer than for 5 layers.

Secondly, we point out that there are no significant changes if we use different optimizers or weight decay. In some cases, the same results are obtained, for example, if we use weight decay 0, 1e-5 or 1e-4, and if we use Adam or AdamW optimizers.

In the paper, we have added the following (section 4.1.1): 'We also trained the number of layers, the optimizer, and the weight decay. However, we did not notice any significant differences when using different optimizers or weight decay.'

And at the end of section 4.2:

'In addition, for the ensemble of models, we noticed further improvements (mainly on the standard deviation) when we added more hidden layers. On an ensemble of 10 trained models, ~0.3 N-unit improvement can be achieved when using 10 hidden layers, compared to 5 layers for the ML model. However, this further increases the training time, which is especially important for the ML method, given that we use a total of 30000 epochs. A higher number of layers is more suitable for BI&ML, where the number of epochs is much smaller.

We point out that the scope of this study is to present ML as an alternative method to grid RO observations. The results obtained herein indicate better performance compared to BI. Considering our results and the additional tests with multiple models, BI&ML brings additional (small) improvement compared to ML methods, with an obvious advantage in terms of a much shorter time to train the model. For future work related to continuous long-term RO-gridded products, an ensemble of models will be trained to also provide the uncertainty related to the models.'

*Table 2: Tuning of number of layers, optimizer, weight decay. Statistics in N-units, (not in manuscript).*

| Nb. of Layers | 1 | 2 | 3 | 4 | 5 | 6 | 7 | 8 | 9 | 10 |
|---|---|---|---|---|---|---|---|---|---|---|
| Av. St.dev | 13.05 | 10.64 | 9.65 | 9.39 | 9.16 | 9.13 | 9.07 | 8.90 | 8.91 | 8.87 |
| Std. St.dev | 0.24 | 0.16 | 0.13 | 0.14 | 0.08 | 0.17 | 0.11 | 0.13 | 0.11 | 0.07 |
| Av. Bias | 0.17 | 0.26 | 0.30 | 0.26 | 0.19 | 0.31 | 0.24 | 0.41 | 0.09 | 0.29 |
| Std. Bias | 0.10 | 0.20 | 0.43 | 0.40 | 0.31 | 0.32 | 0.40 | 0.17 | 0.35 | 0.24 |

| Optimizer | Adam | | | AdamW | | | RMSprop | | | |
|---|---|---|---|---|---|---|---|---|---|---|
| Av. St.dev | 9.16 | | | 9.16 | | | 9.12 | | | |
| Std. St.dev | 0.08 | | | 0.08 | | | 0.08 | | | |
| Av. Bias | 0.19 | | | 0.19 | | | 0.22 | | | |
| Std. Bias | 0.31 | | | 0.31 | | | 0.39 | | | |

| Weight Decay | 0 | | 1e-5 | | 1e-4 | | 1e-3 | | | |
|---|---|---|---|---|---|---|---|---|---|---|
| Av. St.dev | 9.16 | | 9.16 | | 9.16 | | 9.18 | | | |
| Std. St.dev | 0.08 | | 0.08 | | 0.08 | | 0.10 | | | |
| Av. Bias | 0.19 | | 0.19 | | 0.19 | | 0.38 | | | |
| Std. Bias | 0.31 | | 0.31 | | 0.31 | | 0.17 | | | |

**3. Data Preprocessing: The paper does not explicitly mention any preprocessing steps for COSMIC-2 data. It would be insightful to provide details on any preprocessing carried out, as this could significantly influence the model's performance.**

Thank you for this comment. For our application, not much pre-processing is required. The only preprocessing we do is mentioned in the paper, as follows:

i.   COSMIC-2 refractivities are interpolated at isohypsic heights, 2 km, 3 km, 5 km, 8km, 15 km and 20 km. The vertical resolution of RO is relatively high and therefore we do not expect this interpolation to change the distribution of the data. Since the resolution is high, we used a simple linear interpolation. The resulting interpolation error is much smaller than the uncertainty from training different models.

From Figure 1 in the paper, you can see the resulting refractivities at these heights, which appear appropriate to our expectations.

ii.   The input data latitude, longitude and time are standardized.

**4. Data Splitting Strategy: Randomly splitting data into train, valid, and test sets might not be optimal, as these sets could be interrelated (so called 'data leaking'). The suggestion is to have an independent validation set and test set for a more robust evaluation of the proposed models.**

We appreciate this comment and have made revisions to make this clearer in the paper. Although the data are randomly chosen, we point out that the test, validation, and train datasets do not have the same refractivity. By

random we mean that we do not split them according to time or geolocation or refractivity values, however, these datasets do not contain the same points.

In addition, we would like to emphasize that our problem is an interpolation problem and not a prediction problem. We aim to improve the resolution of RO refractivities and produce gridded products. Therefore, it is fine for our application if the training/validation and testing data are not separated epoch-wise, for example, days 1-8 for training and validation and days 9-10 for testing.

In the paper, in Section 4.1 we changed the following sentence:
'As is customary in ML, we randomly split the nature dataset into two segments: 80% of the data for training, 20% for testing.'
to
'As is customary in ML, we randomly split the nature dataset into three segments: 72% of the data for training, 8% for validation and 20% for testing. We point out that the training, validation, and testing datasets do not overlap. By random choice, we mean that we do not split them according to a specific parameter (such as time, geolocation, or refractivity values).'

**5. Code and Data Availability: To enhance the reproducibility and validation of the research, it is recommended that the authors provide complete code and data. This would enable other researchers to replicate the experiments more easily and validate the results effectively.**

Thank you for this suggestion.
- We provide the Matlab routines we have used to read COSMIC-2 refractivity at 2 km height. The data can directly be downloaded from the UCAR website.
- We provide the ML implementation in Python to train and evaluate the mapping of refractivity using ML.
- We provide the mapped refractivities using BI for train and test dataset.
- We provide the COSMIC-2 data (as CSV files), used as input to ML routines.

The codes and data will be available for the scenario at 2 km height when we train COSMIC-2 refractivities. Using the scripts researchers can reproduce the ML and BI&ML results for COSMIC-2 at 2 km directly and can easily adapt the routines to reproduce the results at the other evaluated altitudes.

The section 'Code and data availability' has changed as follows:
'We provide sample routines for the readers to be able to reproduce the results for COSMIC-2 observations at 2 km, https://doi.org/10.3929/ethz-b-000670139. These include sample data (train and test data for refractivity at 2 km) and code implementation (Matlab code to read the COSMIC-2 data at 2 km and Python codes to train and evaluate the ML model). Additional codes associated with this study are available from the corresponding author upon reasonable request. Additional datasets generated during and/or analyzed during the current study are available from the corresponding author upon reasonable request. The COSMIC-2/FORMOSAT-7 data is freely available from UCAR. The ECMWF data are a product of the European Centre for Medium-Range Weather Forecasts (ECMWF) (© ECMWF).'

---

## Author Comment (AC4)

**Reviewer #3:**

**Overall Comments: The work carried out by the authors in the submitted manuscript is of great significance for regularizing the sparsely available GNSS radio occultation (RO) data on a global grid. Such a data may be gainfully used for morphological construction of RO climatology. The technique of machine learning (ML) developed in the work is an advancement over Bayesian Interpolation (BI) – both as standalone model as well as when combined with BI. It is also a timely application of the current global use of AI/ML approaches, especially, with a copious amount of GNSS RO data being available from past and existing satellite missions. The manuscript is clear in its objectives, cogent and systematic in its presentation and well-written in lucid language. Though the authors have liked to restrict their objective to showcase the benefit of ML – as standalone and as combined model with BI, it would have been thorough if some of the advanced ML models such as those based on decision trees viz. random forest, XGBoost etc or using several regressors like stacking regressor were also compared in terms of their statistical metrics. However, this is just a suggestion for future work. I strongly recommend for the manuscript to be accepted for publication in the journal of AMT after incorporation of corrections suggested as minor comments below.**

Dear reviewer, thank you very much for taking the time to review our work, we appreciate your comments and your positivity towards the manuscript. Indeed, in future works, we intend to apply more advanced ML methods to further enhance the results; this work was performed to investigate the potential of ML to construct RO climatologies and compare with the state-of-the-art method Bayesian interpolation.

Following, you can find our answers to all your minor comments.

**Other minor comments:**

**1. Line 107: wetPf2 is not the refractivity rather it is the name assigned to a set of retrieved state parameters using 1dVAR method. It is better to state "analyzed refractivity sourced from wetPf2 files from the data portal ….".**

Thank you for pointing this out. This has been changed accordingly.

**2. Authors to precisely refer in texts (Lines 115-125) to each sub-figures using the assigned alphabets in figure 1. What is the grid size along the zonal and meridional direction chosen for each sub-figures of figure 1.**

Thank you for the suggestion, we have now referred to each sub-figure in the text.
The data plotted in Figure 1 are for the COSMIC-2 distribution, for 10 days in January 2020. These data are plotted for the latitude-longitude coordinates of the COSMIC distribution; it looks like a grid because 40000 data points are plotted together.

**3. Lines 145-150: Authors to mention whether they have accounted for the difference between geopotential fields for interpolated refractivity from ECMWF model forecasts and geometric altitude above mean sea level for COSMIC-2 refractivity before comparing?**

Thank you noticing this. We point out that from the ECMWF data that we have, we also compute geometric altitude above mean sea level using the following formula:
Geometric_alt_above_mean_sea_level = Earth_Radius*geopotential_height/(Earth_radius – geopotential height)
Therefore, we have also used the geometric height above mean sea level for the evaluations of our algorithms applied to ECMWF data, as we did also for the COSMIC-2.

For clarification the part in bold has been added in the text:
'We computed refractivity profiles **(and geometric altitude above mean sea level)** at the times and locations of COSMIC-2 RO soundings….'.

**4. In line 145, what is the reason for not using any prior forecast fields such as 3 hours, 6 hours? Is it availability or any other justifiable reason?**

The main reason is that we want to use ECMWF forecasts as a "nature run" to avoid any effect of assimilation of COSMIC-2 RO data into ECMWF, please see beginning of section 2.2. The main reason that we choose 12 hours is that after 12 hours the effects of the assimilated data in the forecast model is neglectable and the products will be physically consistent. In addition, ECMWF forecasts start at 00:00:00 and 12:00:00, and therefore, we can either have 0-hour forecast (where the assimilated data constrain the atmospheric state) or 12-hour forecast (where the assimilated data do not constrain anymore the atmospheric state).

**5. In line 464: correct the combined model name to BI&ML.**

Thank you for noticing. This has been changed.